Manuscript prepared for Hydrol. Earth Syst. Sci.
with version 2014/09/16 7.15 Copernicus papers of the LATEX class copernicus.cls.
Date: 31 May 2018

# The benefit of seamless forecasts for hydrological predictions over Europe

Fredrik Wetterhall[1] and Francesca Di Giuseppe[1]

[1]European Centre for Medium-range Weather Forecasts, Shinfield Park, Reading ,UK

*Correspondence to:* Fredrik wetterhall: fredrik.wetterhall@ecmwf.int

**Abstract.** Two different systems provide long range forecasts at ECMWF. On the sub-seasonal time scale, ECMWF issues an extended-range ensemble prediction system (ENS-ER) which runs a 46-day forecast integration issued twice weekly. On longer time scales the current seasonal forecasting system (SYS4) produces a 7-month outlook starting from the first of each month. SYS4 uses an older model version and has lower spatial and temporal resolution than ENS-ER, which is issued with the current operational ensemble forecasting system. Given the substantial differences between the ENS-ER and the SYS4 configurations and the difficulties of creating a seamless integration, applications that rely on weather forcing as input such as the European Flood Awareness System (EFAS) often follow the route of the creation of two separate systems for different forecast horizons. This study evaluates the benefit of a seamless integration of the two systems for hydrological applications and shows that the seamless system outperforms SYS4 in terms of skill for the first four weeks, but there both forecasts are biased. The benefit of the new seamless system when compared to the seasonal forecast can be attributed to (1) the use of a more recent model version in the sub-seasonal range (first 46 days) and (2) the much more frequent updates of the meteorological forecast.

## 1 Introduction

ECMWF produces a range of forecasts, among them a 10 day deterministic high resolution forecast (HRES) and a lower resolution 15-day 51 member ensemble prediction system (ENS) that is extended to 46 days twice weekly (Mondays and Thursdays at 00UTC; Vitart et al. 2008). In this paper we refer to the extended ENS as ENS-ER. On longer time ranges ECMWF issues a seasonal ensemble forecast system (SYS4), operational since November 2011. SYS4 issues a 7-month prediction (extended to 13 months four times a year) once every month (Molteni et al., 2011). The ENS-ER

forecast system benefits from frequent updates of the model physics and data assimilation system (Vitart et al., 2008). ECMWF releases official model updates on average 2-3 times a year which typically include new improved schemes for physical processes, better use of observations and their
assimilation and sometimes increase in model resolution. The seasonal forecast has a lower resolution, is an older model version than ENS-ER and is also updated much less frequently. This implies that the skill of the seasonal forecasting system is lower relative to ENS-ER in the overlapping first six weeks.

Applications that use numerical weather predictions as forcing, such as the operational European
Flood Awareness System (EFAS; Thielen et al. 2009; Bartholmes et al. 2009; Smith et al. 2016), are often designed for a specific purpose. EFAS has since the start focused on early warning of floods in the medium-range forecast horizon, typically up to 15 days. Recently, a seasonal hydrological outlook forced by SYS4 was implemented operationally with a lead-time of 7 months (Arnal et al., 2018).

This extension to the monthly and seasonal scales is potentially very useful in order to; (i) produce products which extend the previous forecast horizon; (ii) benefit from hindcasts for pre- and post-processing to produce output of higher quality (e.g. model based return periods); and (iii) design completely new early warning frameworks complementing the existing ones. The extended lead time provided by running EFAS forced by weather prediction across different time scales could po-
tentially provide added benefit in terms of very early planning, for example for agriculture, energy Bazile et al. (2017) and transport sectors Meißner et al. (2017) as well as water resources management Sene et al. (2018). Such a forecast system would be a first step to close the identified gap between hydrological forecasts on the medium (up to 15 days) and seasonal range (White et al., 2017). These extended range systems may not be able to capture extremes of short-lived events like
floods, but they are able to detect anomalous conditions on longer lead times, such as low flows (Meißner et al., 2017) and droughts (Dutra et al., 2014).

The concept of seamless forecasts was first introduced by Palmer and Webster (1993). Palmer et al. (2008) formally expanded the idea showing how short-lived phenomena under certain conditions may persist and increase predictability at longer time scales. Since then the concept of a unified
or seamless framework for weather and climate prediction has been vastly debated (Hurrell et al., 2009; Brunet et al., 2010). However as noticed by Hoskins (2013) in his seminal paper, while "the atmosphere knows no barriers in time-scales", often model implementation is segmented for practical reasons. Still, major efforts have been made to create unified systems. Indeed, the ENS-ER was the first attempt to create a seamless extension of the ECMWF medium-range forecast to the monthly
scales(Vitart et al., 2008). Similarly, the UK Met Office has in the past twenty-five years worked to create a unified model that could work across all time scales (Brown et al., 2012). Also the climate community has moved in the same direction. For example, the EC-Earth project shows that a bridge can be made between weather, seasonal forecasting and beyond (Hazeleger et al., 2010, 2012).

The latter projects went all the way to create new systems starting from existing components and were therefore costly and time demanding. In contrast, a practical and simpler approach could be taken. The seamless idea could be translated into a concatenation of "the best" forecast at each lead-time. The clear advantage of this off-the-shelf seamless prediction conversion is that it utilizes products that are already available and operational, thereby avoiding the complications of new developments, while at the same time generating forecast products to meet different types of users (Pappenberger et al., 2013). There is however an underlying complexity in this simplification; the difference in design between the various forecasting systems makes the concatenation not entirely straight-forward. The forecasting systems are related since they are from different generations of the same model development, however they have non-matching temporal and spatial resolutions, different hindcast span and different ensemble sizes. One important consequence of this is that the more frequent updates to the extended range compared to the seasonal forecasting system at ECMWF causes the model errors from the two systems to diverge over time, and only closing this gap when the seasonal system is updated to a newer model version (Di Giuseppe et al., 2013). Then model outputs either need to be bias-corrected to be useful forcing to drive sectoral models such as EFAS, or that final products should be provided in terms of anomalies calculated against the model climate, taken into consideration the bias of the seamless forecast system. In both cases the seamless system needs to account for the use of the hindcast dataset and the application of some bias correction algorithm. In return, the advantage is in the gain in skill and the extension of the lead-time.

In this work the benefit of a seamless hydro-meteorological system was tested for a span of time ranges from 1 week to 6 months for stream flow forecasts over the European domain using the EFAS system. The aim was to test whether integrating medium-range forecasts with seasonal prediction contributes to enhance hydrological predictability on the seasonal scale. Specifically, the questions addressed were: What is the gain in terms of hydrological forecasting of using a more recent model version in the first 46 days provided by the use of the ENS-ER? What is the skill gain provided by having more frequent forecast updates?

## 2 Methods

### 2.1 Hydrological model system

The hydrometeorological system used in this study was the European Flood Awareness System (EFAS Thielen et al. 2009; Bartholmes et al. 2009; Smith et al. 2016). EFAS is an operational early warning system covering most of the European domain and has been run operationally since October 2012 as part of the COPERNICUS Emergency Management Service (CEMS). The hydrological component of EFAS is the distributed rainfall-runoff model LISFLOOD (De Roo et al., 2000; Van Der Knijff et al., 2010; Burek et al., 2013). LISFLOOD calculates the main hydrological processes on sub-daily and daily time-scales that generate runoff for each grid cell. In the operational setup

EFAS covers most of Europe on a 5x5 km equal-area grid. The runoff is transformed through a routing scheme to estimate the river discharge at each grid cell along the river network. The routing scheme also takes into account water retention in lakes and reservoirs. This study will concentrate on the forecast of river discharge at the outlets of the sub basins of the river network that were used for calibration of the current EFAS system (Smith et al., 2016; Zajac et al., 2013). The total number of outlets used were 679, and they represent river basins of all sizes and characteristics across the EFAS domain.

In its operational implementation the latest calibration (referred to as tuning in the NWP nomenclature) of LISFLOOD used an observational dataset of meteorological forcing data (precipitation and temperature) and observed discharge covering the model domain over the period 1990-2013 (Smith et al., 2016; Zajac et al., 2013). The meteorological dataset comprises more than 5000 synoptic stations that have been interpolated to a 5x5 km Lambert azimuthal equal-area projection (Ntegeka et al., 2013). The high resolution gridded observation of precipitation and temperature were used for the calibration of LISFLOOD. The observational meteorological dataset was also used to generate a reference modeled climatology of discharge (hereafter called water balance, WB) which is used as; (i) initial conditions for the operational forecast and hindcasts and (ii) reference model run to assess the performance of the forecasts. Using the WB run as proxy observation simplifies the interpretation of the skill scores as it avoids the complication of having to assess the bias against observed discharge. The purpose of this paper is rather to assess the skill of the two forecasts used for forecasts rather than the total skill of the forecasting system.

**2.2 Seamless integration of meteorological forcing data**

Every Monday and Thursday ECMWF issues an extended-range ensemble forecast (ENS-ER) by continuing the integration time beyond day 15 up to day 46, with a lower-resolution model (Figure 1, Table 1). Each ENS-ER integration comes with an 11-member hindcast set produced for the same dates as the forecast date over the previous 20 years. This hindcast set provides identical integrations as the current operational forecast with the difference that ERA-Interim reanalysis (ERAI; Dee et al. (2011)) and ERAI land reanalysis (Balsamo et al., 2015) are used to provide the initial conditions, whereas the operational ensemble forecast uses the operational analysis. The hindcast data together with observations can be used in many applications, for example to calibrate the forecast in an operational setting (Di Giuseppe et al., 2013).

The operational seasonal forecast (SYS4) issues a new forecast at the beginning of each month with a lead-time up to 7 months, four times a year extended to 13 months (Figure 2). SYS4 has a hindcast consisting of 30 years started at each month and consisting of 15 members. The new seamless forecasting system (hereafter called SEAM) was created by concatenating each ENS-ER ensemble member with a randomly selected SYS4 ensemble member at day 46, which is the last day

of the ENS-ER (Figure 2). SEAM benefits from the frequents updates of the ENS-ER and has the seven months horizon of the seasonal system.

Since the two systems have different resolutions (table 1) the horizontal resolution was homogenized to the 5x5 km equal-area grid through a mass-conservative interpolation for precipitation and a bilinear for temperature before it was used as input to the hydrological model in EFAS. The mass-conservative interpolation summarizes the partial contribution of the meteorological input fields onto the LISFLOOD grid. The time step was reduced to daily by averaging (accumulating for precipitation and evapotranspiration) the three hourly outputs of the ENS-ER and the six hourly outputs of SYS-4. Since the ENS-ER has a reduced hindcast (20 years) and number of members (11), SEAM has the same number of members and hindcast period. Note that in real-time mode, a full 51-member SEAM is possible. The technical details of the forecast and the hindcast used in this experiment are presented in table 1. For simplicity SYS4, and SEAM will from now on refer to the full hydro-meteorological model chain and not only the meteorological forcing for the remainder of this paper.

### 2.3 Experimental set-up

This study focuses on the performance of SYS4 and SEAM over the hindcasts of the operational forecast. The hindcasts starting from 2015-05-14 (the first available date with 11-member hindcast for ENS-ER) to 2016-06-02 were used used as input to the full EFAS modeling chain. As described above, the hindcasts are the reforecasts over the previous 20 years and is produced for each individual run of the ENS-ER. This provided 13 monthly starting dates for SYS4 and 111 biweekly starting dates for SEAM with a corresponding hindcast set covering all seasons over the previous 20-year period, each with 15 and 11 ensemble members respectively (Fig. 1). The output was averaged to weekly means before the skill score analysis. Since the starting dates of the SEAM and SYS4 were not always in sync (the starting date of the SYS4 integrations are only sometimes on a Monday or Thursday), it is impossible to do a completely like-for like comparison since the validation periods would be slightly different. However, this error will be random and given the sample size (260 and 2220) it was not considered to have a big impact on the results.

SEAM was validated against the runs with SYS4 to assess the added value of the merged forecast. Further, both model systems were compared against a climatological benchmark simulation (hereafter called CLIM). CLIM was constructed by forcing the LISFLOOD with 15 randomly selected time series of observed meteorological forcing from the period 1990-2014, excluding the modeled year. CLIM has the advantage of having the same initial conditions as the SYS4 and SEAM hindcasts, but has no expected predictive skill beyond the horizon of the initial conditions. The advantage of CLIM is that in theory it has near perfect reliability with regards to the WB runs since it is produced with the same unbiased forcing data. It should therefore score better or equal as the hindcasts as predictor on time ranges beyond their respective limits of predictability.

## 2.4 Score metrics

The performance of the two forecast systems was compared against the WB run at the 679 sub basin outlets using deterministic and probabilistic scores. WB is treated as a proxy for observations in the evaluation. The scores used were the continuous ranked probability score (CRPS; Hersbach 2000), mean relative error (MRE) and forecast reliability through and attributes diagram. All scores were calculated for SYS4 and SEAM over the hindcast period. CRPS is a common tool to validate probabilistic forecasts and can been seen as generalization of the mean absolute error to the probabilistic realm of ensemble forecasts. It is defined as:

$$CRPS = \frac{1}{N} \sum_{t=1}^{N} \int_{-\inf}^{+\inf} \left[ F_t(x(n)) - H_t(x(n) - x_0)^2 \right] dx \tag{1}$$

where $x(n)$ is the forecast at time step $t$ of $N$ number of forecasts and $x_0$ is the observed value (WB). The CRPS is the continuous extension of the ranked probability score (RPS), where $F(x)$ is the cumulative distribution function (CDF) $F(x) = p(X - x)$ and $H(x - x_0)$ is the Heaviside function, which has the value 0 when $x - x_0 < 0$ and 1 otherwise.

The CRPS compares the cumulative probability distribution of the discharge forecasted by the ensemble forecast system to an observation. It is sensitive to the mean forecast biases as well as the spread of the ensemble. Since the SEAM has 11 members and SYS4 and CLIM has 15 members in the hindcast, the CRPS are not directly comparable. Ferro et al. (2008) showed that for two ensemble distributions with different ensemble sizes, $M$ and $m$, the unbiased estimate for $CRPS_M$ based on CRPS calculated from the ensemble size $m$ is:

$$CRPS_M = CRPS_m - \frac{M-m}{2Mmn} \sum_{t=1}^{n} \Delta_t \tag{2}$$

where

$$\Delta_t = \frac{1}{m(m-1)} \sum_{i \neq j} |X_{t,i} - X_{t,j}| \tag{3}$$

is Gini's mean difference of ensemble members $[X_{t,1}, ..., Xt, m]$ at time $t$. From the CRPS a skill score (CRPSS) can be derived by comparing CRPS of the verified forecast against a reference forecast.

$$SS_{CRPS} = 1 - \frac{CRPS_{fc}}{CRPS_{rf}} \tag{4}$$

Mean relative error (MRE) was measured as the forecast bias in comparison with WB normalised with WB, here defined as:

$$MRE = \frac{1}{n} \sum_{t=1}^{N} \frac{x_o - \overline{x}(n)}{x_o} \tag{5}$$

where $x_o$ denotes the observed value (WB) and $\overline{x}(n)$ denotes the forecasted ensemble mean at time $t$.

The reliability was assessed through an attributes diagram, where the forecast probability of exceeding a certain threshold is compared with observed frequencies Hsu and Murphy (1986); Weisheimer and Palmer (2014). The forecast reliability was evaluated for the 10, 50 and 90 percentiles of observed discharge at each outlet.

## 3    Results and discussion

### 3.1    Overall forecast skill

The forecast skill gain provided by SEAM with respect to SYS4 is mostly concentrated to the first six weeks (Figure 3,a) when the forcing data are from the ENS-ER. The difference in CRPSS is 0.6 at week one, which then decreases to 0.1 by week six. All points used in the validation show a gain in skill up until week three, then some points show a benefit of using the SYS4 instead of

SEAM. However, in some catchments there is skill up further than eight weeks. The overall better performance of SEAM with respect to SYS4 is partly because of the use of a more recent model version and partly because of the more frequent update of the atmospheric and hydrological initial conditions. It is possible to disentangle the relative contributions between these two factors by only considering a reduced number of starting dates for the SEAM forecast; i.e dates that are the closest

to the SYS4 starting dates (Figure 3,b). This reduced statistic provides a measure of the expected contribution of *only* employing a newer model cycle in the first weeks while both simulations benefits from the same hydrological initialization. In this case the skill gain in CPRS reduces to between 0 and 0.4 (median 0.2) against SYS4 for the first week, reducing to neutral around week four. Therefore the most relevant gain comes from the more frequent initializations of the hydrological model.

To put these increments into context we also look at the improvement in skill of the two system SYS4 and SEAM against the CLIM benchmark forecast (Figure 3c-d). The gain from having an improved initial conditions in SEAM is similar in comparison with CLIM (Figure 3c) as with SYS4 (Figure 3a) in the first week, but the skill deteriorates quicker and the median CRPSS is negative after 5 weeks. Without the increase in skill due to the advantage in the better initial conditions, SEAM

still shows a gain against the CLIM forecast with a CRPSS of 0.4 for the first week, although the spread is quite large (Figure 3d). Also SYS4 shows an increase of skill against the CLIM forecast. Both forecasts are less skillful than CLIM for most river points after week four. It can also be noted

that SEAM has a higher spread than SYS4 on longer lead times even though they are forced with the same data from day 47 and onwards. An explanation can be that the ensembles from the two meteorological forecasts are not matched member by member in terms of their relative deviation from the mean, for example matching members from each distribution according to their wetness. If two extreme driving forecasts from the two meteorological forecasts are combined it can lead to members that are further away from the ensemble mean than when only one driving forecast is used.

## 3.2 Geographical variation of forecast skill

The geographical distribution of skill gain provided by the SEAM and SYS4 prediction reveals a coherent picture with good scores against the CLIM run over most of Europe (Figure 4 a-b). The gain in the figure is expressed as a difference in the number of weeks into the forecast needed for the CRPSS to drop below zero (i.e. there is no skill in the forecast in comparison with CLIM), which gives an indication of the expected time gain in terms of information provided by the forecast against the reference forecast. Both SYS4 and SEAM are better than CLIM, and SEAM has higher skill than SYS4 for most of Europe. There is a small negative effect over the Alps, southeastern Europe and northern Finland (Figure 4d). The performance of the operational EFAS in these regions is generally poor, which is caused by the difficulty of having good observations of precipitation in high altitude stations and the atmospheric models difficulty in resolving steep orography (Alfieri et al., 2014)."The snow accumulation and snowmelt are further divided into three elevation zones within a grip in LISFLOOD to better account for orographic effects in mountainous regions. However, this increase in sub grid resolution is not likely to be high enough to capture the snow variability during the snow accumulation and snowmelt in mountainous regions. Further, precipitation forecasts have documented biases in steep orography ((Haiden et al., 2014)).

Another interesting aspect to showcase is the relevance of more frequent model version updates is the overall improvement on river discharge for all stations in proximity to the western coasts. This can be attributed to recent developments of the precipitation model scheme, for example a new diagnostic closure introduced in the convection scheme (Bechtold et al., 2014) and a new parameterization of precipitation formation (Haiden et al., 2014).

## 3.3 Bias and reliability

The relatively sharp decline in CRPSS can to some extent be explained by the negative bias (too wet forecast) for both SEAM and SYS4 forecast (Figure 5). SEAM has lower bias than SYS4, also when the analysis is confined to the first few weeks (Figure 5b). The slightly better bias in SEAM disappears quickly after the merge (week 7). The bias of the forecast is not spatially consistent, it is generally larger west- and mid-Europe (Figure 6). The figure shows the bias for SEAM (a-c) but the pattern is similar for SYS4. SEAM has generally a smaller bias than SYS4 (Figure 6d). SYS4 has lower bias south of the alps, where it is also performs better than SEAM.

Reliability of a forecast is terms of its usefulness for decision making. A reliable forecast can be trusted to predict the correct probability of certain events, regardless of the accuracy. An unreliable forecast is in practice of no use and can lead to poor decisions Weisheimer and Palmer (2014). Both forecast systems are over-confident when it comes to predicting the median flow, which can be attributed to an underestimation of the ensemble spread (Figure 7. The results are comparable to a previous study of 2 m temperature and precipitation over Europe with SYS4 Weisheimer and Palmer (2014). The reliability with regards to low flows (dashed line, figure 7) indicates an over-prediction of low flows, which can be explained by a the wet bias of the both systems causing an over-estimation of the low flows. SYS4 is performing better than SEAM in this regard. The high flows are generally underestimated by both systems, but SEAM performs slightly better than SYS4 (dotted line, figure 7). The skill of the forecasts from any of the system could be potentially higher by performing a bias correction either of the atmospheric input and/or of the discharge. However in this paper we concentrate on the difference in skills provided by the various configurations and no bias correction has been applied.

### 3.4 Added value of the seamless forecast

Even though the increase in the overall skill provided by the SEAM in comparison with SYS4 is noticeable, the justification for its use in an operational context also depends on the actionable time gain in a response situation. More frequent forecast updates could potentially be useful in decision making. As an example we analyze the predicted stream flow for the Rhine river at a station just upstreams Cologne, Germany, during the European heat wave in the summer of 2003. It was an exceptional meteorological event which combined significant precipitation deficits with record-setting high temperatures (García-Herrera et al., 2010). At its peak in August, extremely low discharge levels of rivers were reported in large parts of Europe. Economic losses where huge in many primary economic sectors including transportation (Ciais et al., 2005). For several months inland navigation was heavily impaired and the major European transport routes in the Danube and Rhine basins ceased completely (Jonkeren et al., 2008).

Despite the fact that 2003 conditions were extreme from the meteorological point of view, the upcoming deficit in precipitation and the high temperatures were well predicted by the ECMWF seasonal systems operational at that time (System-3; Weisheimer et al. 2011). The good predictability of the event is confirmed by the low discharge prediction provided by SYS4 at the Rhein upstreams of Cologne (figure 8). More then 30 % of the ensemble members forecast extreme low-flow conditions. In fact the observed discharge confirms that the river flow on two separate occasions, event one on August 17-27 and event two September 18-28 2003, went below the 3% percentile of its climatological value for the season (figure 8). While most of SYS4 ensemble members mark the extreme condition three to four weeks ahead, there is no information of the recovery period observed between event one and two in the forecast starting the first of August. SYS4 predicts a swift

recovery back to normal conditions on the forecast issued 1 September. A more detailed picture of
this intermediate recovery is instead conveyed by the seamless system. Thanks to the more frequent
updates, the temporary increase in river flow is correctly picked-up giving a potential advantage of
two to three weeks for planning actions. SYS4 does indicate the second low flow with a longer lead
time than SEAM. However, SYS4 misses the timing of the event.

Even if this was a good forecast for SYS4, the information it provides is more informative
(anomaly condition) than "actionable" (White et al., 2017). In the above example, a decision maker
would have to make a decision based on a forecast that was issued 2.5 weeks earlier, which would
inherently make the decision more uncertain if you only had the seasonal forecast. With the seam-
less system available a decision maker would gain the same early indication of a hazardous event
and also have the benefit of frequent updates. In this particular case, the SEAM forecast for the first
event was more unstable for some ensemble members, but in general the event was well captured
(Figure 8). The SEAM could also correctly capture the recovery with higher water levels between the
extreme low flow events. The onset of the second low period was correctly modeled by the SEAM
system, whereas the timing of the low flow was missed by SYS4. It should be said that using other
less extreme thresholds (<90 and <95 percentiles) even further strengthened the case for using the
SEAM.

## 4 Conclusions

This study compared a set of hydrological hindcast experiments over the European domain with two
meteorological forcings; ECMWF's seasonal forecasting system (SYS4) and a merged system of
ECMWF extended range forecast and seasonal forecast system (SEAM). The latter showed a bet-
ter overall skill and lower bias over most areas in Europe with lead times up to seven weeks. This
increase in skill could be attributed to better initial conditions of the hydrological and meteorolog-
ical model as well as a better atmospheric model version in SEAM. In some areas, particularly in
the Alps and northern Finland, the seasonal forecast outperformed the merged forecast. However,
in these areas the predictability the hydrological model is generally poor which makes these results
quite uncertain. Given that the skill in the sub-seasonal range over Europe is in the range of the
extended-range ensemble forecast would motivate to use the ENS-ER instead of SYS-4 for hydrom-
eteorological predictions.

Still, there is an added benefit of using a seamless forecast over the extended range due to the
extension of forecast horizon for the early detection of upcoming anomalous conditions. Indeed, as
an example this study also highlighted the potential for the use of a sub-seasonal to seasonal forecast
in the case of an extreme low-flow situation in the River Rhine. The higher frequency and skill of
SEAM has the advantage of being a more "actionable" forecast than seasonal forecasts, given that a
decision maker would be able to make use of the extra information. Care should be taken when using

the forecasts in decision making since the reliability over Europe is "marginally useful" Weisheimer and Palmer (2014). It is therefore important to assess the reliability and skill of SEAM at the location it is to be implemented over the season of interest.

Future work with the seamless forecasting system is to further explore the limits of predictability, reliability and bias to assess the strengths and limitations of the current setup. The assumption that the forecasts can be randomly concatenated would also need to be tested against a system where the forecasts are matched according to their respective climatology. Bias correction of the forecasts might be a necessity, and the advantage of the extended-range and seasonal forecasts from ECMWF is that the availability of hindcasts which are enables just that.

*Acknowledgements.* This paper was financed through the Framework service contract for operating the EFAS computational centre in support of to the Copernicus Emergency Management Service (EMS)/Early Warning Systems (EWS) 198702. The authors would like to acknowledge Blazej Krzeminski for setting up the computational framework and Florian Pappenberger for the discussions regarding the seamless forecast system. We would also like to thank Kean Foster, Bastian Klein and Mike Hardeker for useful comments on the discussion paper.

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

**Table 1.** technical details of the forecast and the hindcast used in this paper.

| System | T Res | Spatial Res | Horizon | Ens size | Issue frequency | Hindcast set | Hindcast Ens size |
|---|---|---|---|---|---|---|---|
| ENS-ER | 3h/6h | 18/36 km[1] | 46 days | 51 | Twice weekly | 20 years | 11 members |
| SYS4 | 6h | 80 km | 7/13 months | 51 | Monthly | 30 years | 15/51 members |
| SEAM | 6h | 5 km | 6 months | 51 | Twice weekly | 20 years | 11 members |

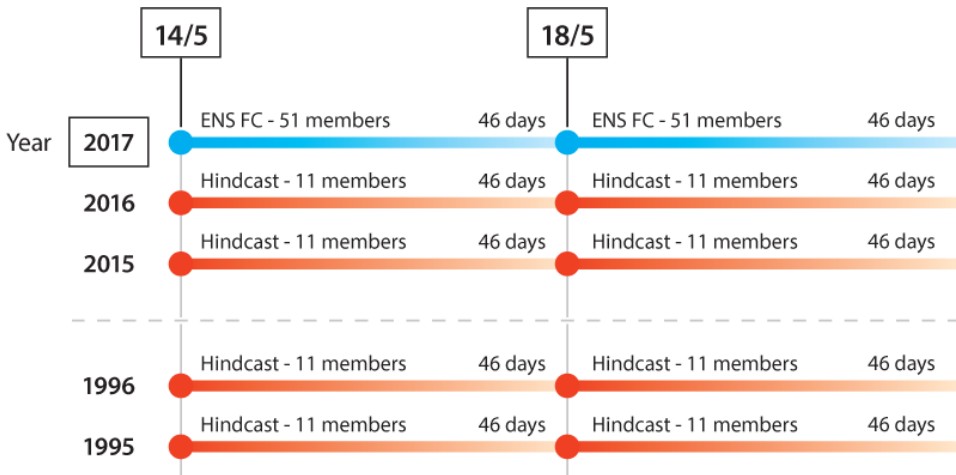

**Figure 1.** Schematic overview of the operational ECMWF ensemble forecast for the extended range and its associated hindcast. The hindcasts consists of a reduced ensemble forecast (11 members) with the same starting date of year as the current forecast, but run for the previous 20 years.

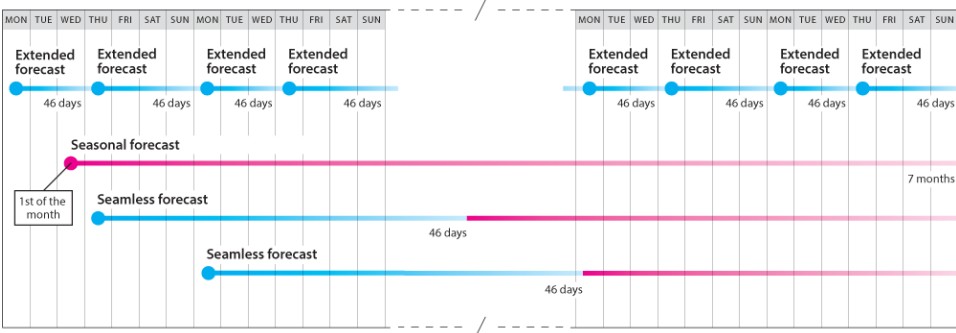

**Figure 2.** Schematic overview of the seasonal, extended-range forecast and merged systems. The Extended forecast is issued every Monday and Thursday and extends up until 46 days, the seasonal forecasts is issued on the first of each month and extends up until 7 months (13 months in February, May, August and November). The merged forecasts concatenates the latest extended forecast with the latest seasonal forecast.

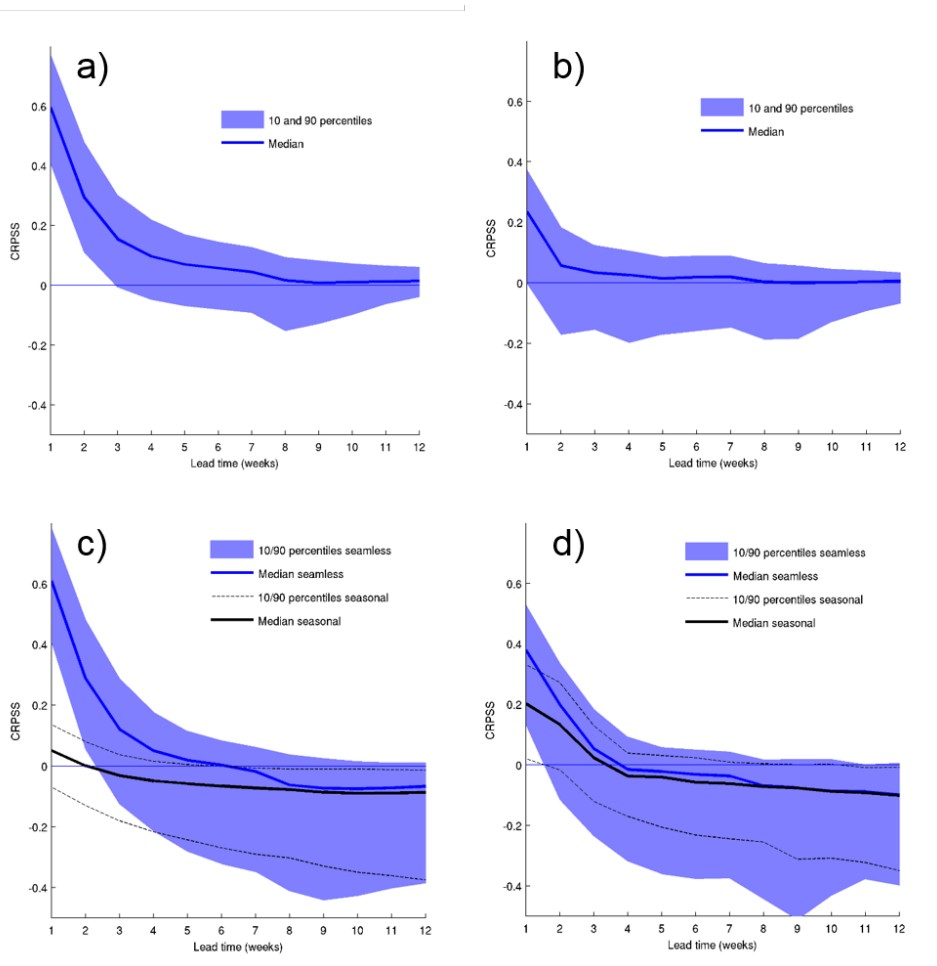

**Figure 3.** Continuous ranked probability skill score (CRPSS) for a) merged forecast against seasonal forecast for all start dates evaluated over the 679 basin outlet points; b) as in a) but only for the first merged forecast of each month; c) merged forecast against climatology for all lead times in blue and d) as in c) but for the first merged forecast in the month. The shaded blue area denotes the 10-90 percentile of the CRPSS and the blue line the median. The black solid (dotted) lines in figure c and d denote the mean and 10-90th percentile of the CRPSS of the seasonal against the climatological forecast.

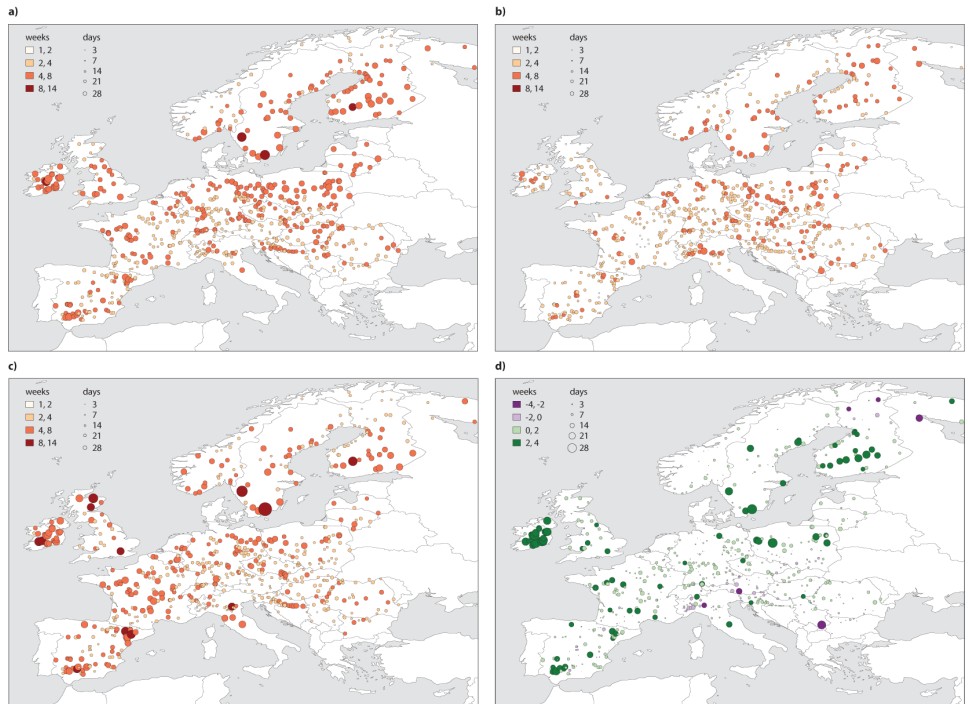

**Figure 4.** The number of weeks (days) before the CRPSS goes below zero using only the first forecast of the month for a) SEAM against CLIM; b) SYS4 against CLIM c) SEAM against SYS4; and d) difference between SEAM against CLIM and SYS4 against CLIM. The dimension of the circles is proportional to the number of days while the color scale refers to the number of weeks. The size and colour of the circles are therefore showing the same information, and are both added for clarity.

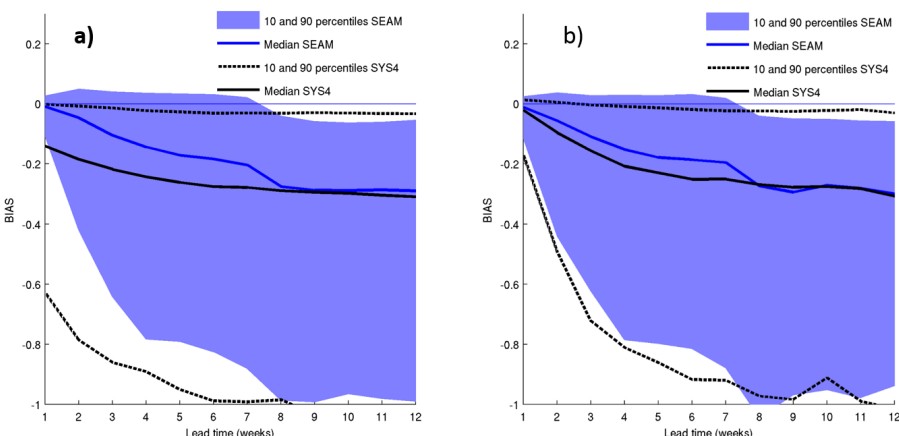

**Figure 5.** Mean relative error over all outlet points as a functionfunction of lead time in weeks for a) all starting dates of the forecasts and b) for the starting dates close to the beginning of the months. Negative values denote that the forecast is too wet in comparison with the CLIM run. The SEAM (SYS4) forecast is in blue (black) where the solid line denotes the median and the filled area (area between dotted lines) denote the 10th to 90th percentile.

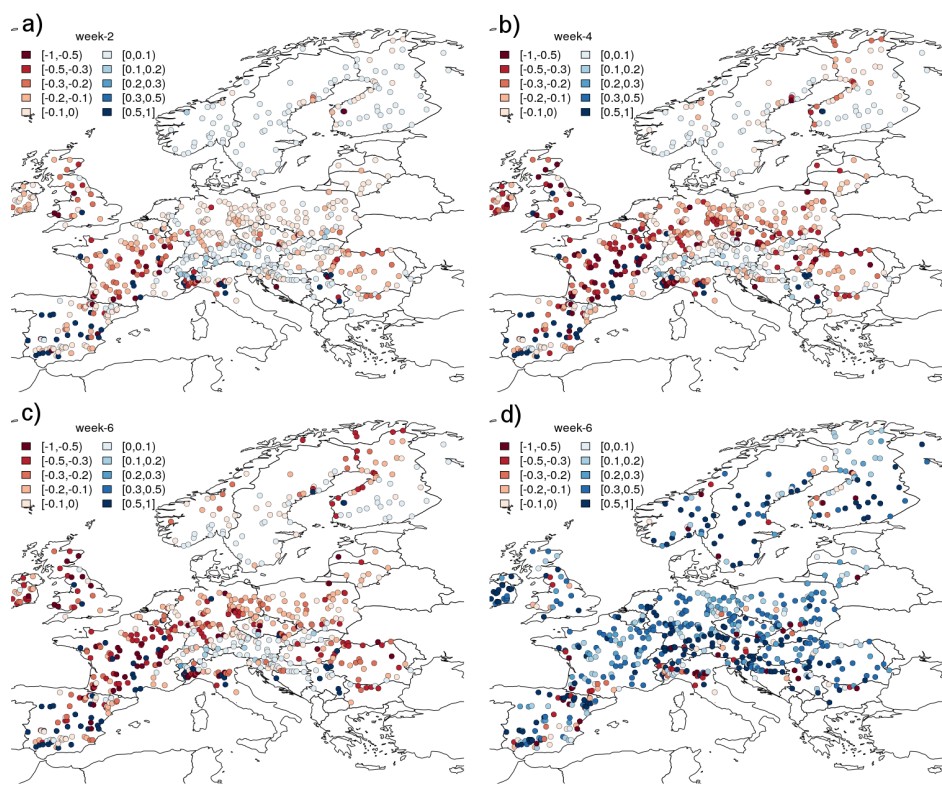

**Figure 6.** Mean relative error for each of the outlet points for the SEAM forecast over the outlet points for a) week 2, b) week 4 and c) week 6. Red indicates where the forecast is too wet, and blue where it is too dry. Figure d) shows the difference in absolute error between SEAM and SYS4, where blue (red) denotes points where SEAM has a smaller (larger) MAE than SYS4.

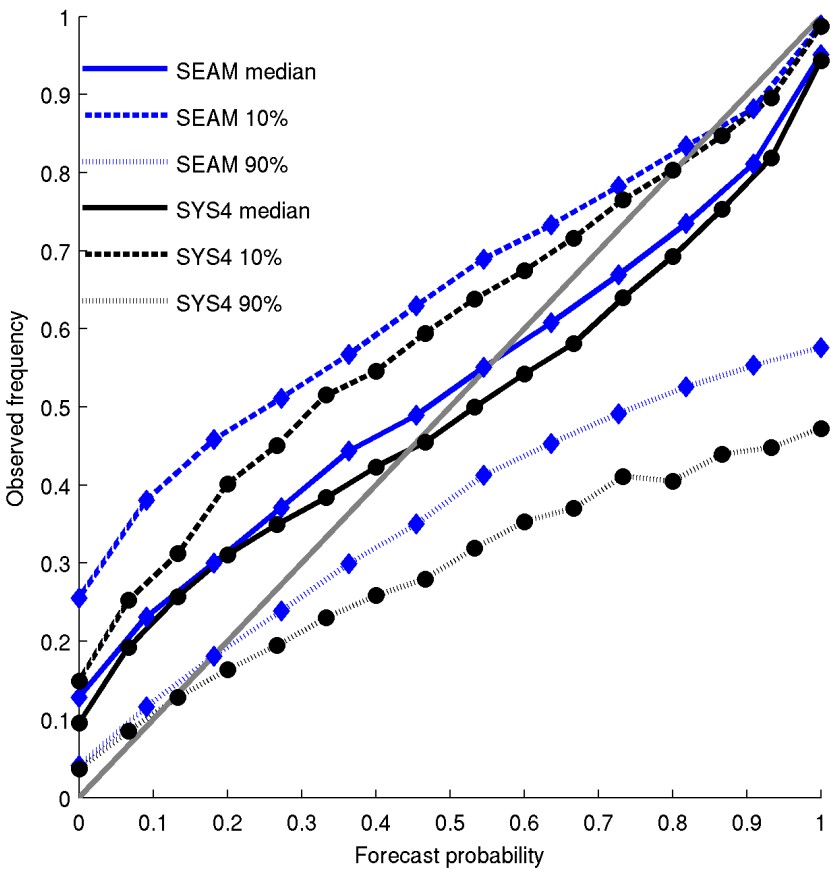

**Figure 7.** Reliability diagram for SEAM (blue) and SYS4 (black) for week 4 for all outlet points. The solid lines indicate the reliability for the median of observed discharge, the dashed (dotted) lines the forecast reliability for the 10th (90th) percentiles of observed discharge.

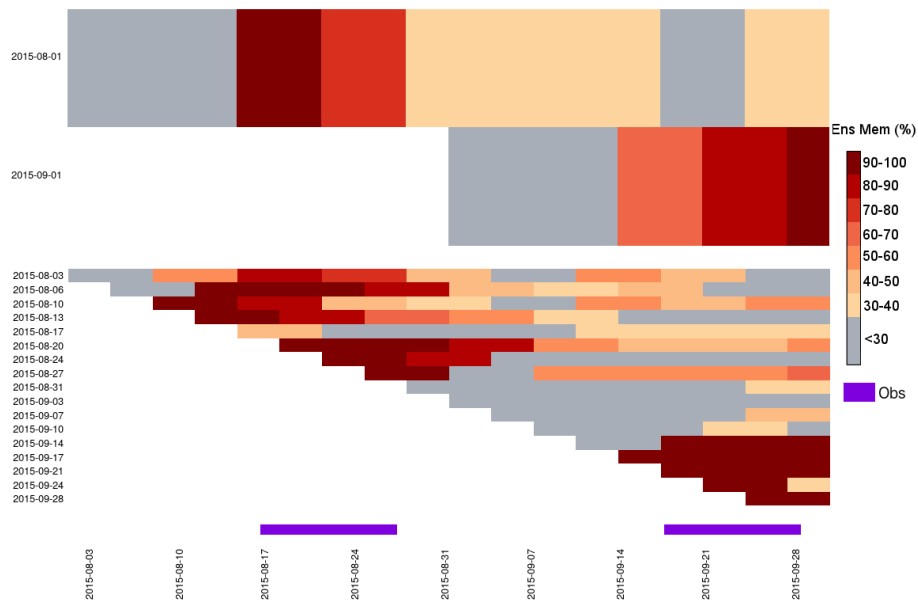

**Figure 8.** Percentage of ensemble members predicting low flow anomaly (< 97%) on the Rhine river north of Cologne for summer 2003. The two starting dates in August and September from SYS4 are compared to the 17 starting dates of the seamless forecasting system. In two separate events the discharge was recorded below the 97 % percentile, event 1 on 17-27 of August and event 2 on 18-28 of September 2003.