# Peer review of "The benefit of seamless forecasts for hydrological predictions over Europe"

_Hydrology and Earth System Sciences, 2017_

## Referee Comment (RC1) · K. Foster (Referee) · 10 Oct 2017

Summary:

In this paper the authors evaluate the added benefit of using a seamless integration (SEAM) of the outputs from ECMWF extended-range ensemble prediction system (ENS-ER) and the ECMWF system 4 seasonal forecast system (SYS4) for hydrological applications. The added benefit from this approach is evaluated by comparing the continuous rank probability scores for the outputs from the hydrological model LISFLOOD forced by SYS4, SEAM, and a climatological ensemble (CLIM) over the hindcast period.

The authors find that hydrological hindcasts made using SEAM show better skill, over those made using SYS4, for much of Europe with lead times up to seven weeks. In some areas like the parts of Alps and northern Finland the reverse was true; however these results are uncertain due to the general poor performance of LISFLOOD in these regions. They argue that the increased skill can be attributed to the better initial conditions of the hydrological and meteorological conditions (models are initialised biweekly as opposed to once per month for SYS4) as well as the use of a better atmospheric model in SEAM (the atmospheric model used in SYS4 is locked at the initial version released with system 4 while the one used in ENS-ER is updated regularly). They conclude that the use of SEAM for hydrological forecasting at the seasonal scale has an added value for decision makers given the higher frequency of updates and improved skill, especially at the sub seasonal scale, making the forecasts more actionable.

The topic of this paper is of great interest at the moment considering the increased focus on forecasting at the sub seasonal to seasonal scales in recent years. Although the concept is not new this paper is the first, that I am aware of, that makes an attempt to evaluate a system that utilises current 'off-the-shelf' operational products. The paper is well written with a good structure and generally clearly formulated, the methods are scientifically sound, and the results are interesting. Additionally, the research presented in this paper is very relevant to the topic of this special issue. In my opinion, the manuscript has a lot of potential for publication in this HESS special issue. However the authors need to clarify some points and revise some statements so that the paper is more easily understood.

General comments:

1) I feel that it is not clear for what periods the study was performed, something which has a bearing on the quality of the results. The authors state that (P5, L132-L135)

"This study focuses on the performance of SYS4 and SEAM over the hindcast period of the operational forecast with a sequence of starting dates over the period 2015-05-14

(the first available date with 11-member hindcast for ENS-ER) to 2016-06-02 producing daily output time series of discharge over the 20-year hindcast period."

The first part of the sentence suggests that the evaluation period is between the dates 2015-05-14 and 2016-06-02 yet the second part says that the hindcast period has a length of 20 years. The next line has a similar mixed message. From the paper I get the general impression that the evaluation is done for the 20 year period so I assume that the issue is to do with how section 2.3 is worded. This should be addressed as there is some confusion in the way that the paragraph (p5, L231-L238) explains it. Further it has implications on the robustness of the results, should the evaluation period be just the 13 months between the aforementioned dates this would give a limited data sample from which to draw the wider reaching conclusions made by the authors. How can the authors know whether the performance of the different approaches during that period was typical of their general performance?

2) The results show that SEAM has skill over SYS4 in the first 3-8 weeks (Figure2b), mostly concentrated in the first 6 weeks. This would imply that there may be a benefit of merging the two meteorological forecasts before day 46. Did the authors consider this and if not why?

Specific comments:

P2, L26: "TSYS4 is also..." I assume that this is a typo and should read, "SYS4 is also..."

P7, L224-L225: Although this line is factually correct it appears to contradict the preceding ones. The reader is being told how the low flows during this period caused substantial economic losses due to it affecting inland navigation in the Danube and Rhine basins only then to be told that navigations are regulated during high flows and not low flows. I suggest rewording this or removing this sentence to remove the perceived contradiction or removing this line altogether as it does not add anything significant to the discussion.

P8, L249-L250: The second part of this line is awkward to read and should be rephrased.

P11, L350-L35: I think the reference is - Pappenberger, F., Wetterhall, F., Dutra, E., Di Giuseppe, F., Bogner, K., Alfieri, L., and Cloke, H. L.: Seamless forecasting of extreme events on a global scale, pp. 3–10, Proceedings of H01, IAHS-IAPSO-IASPEI Assembly, Gothenburg, Sweden, July 2013 (IAHS Publ. 359, 2013)

P16, Caption to figure 3: The last line states, "The dimension of the circles is proportional to the number of days while the color scale refers to progressive weeks." What do the authors mean by number of days?

---

## Referee Comment (RC2) · B. Klein (Referee) · 11 Oct 2017

The manuscript shows the development and the skill of a seamless hydrological forecasting system from sub-seasonal to seasonal scales. Meteorological forecasts from ENS extended (day 1 – 46) and SYS4 (47 to month 7) are merged by randomly selecting ensemble members of SYS4 after ENS extended ends. The skill analysis shows that most of the skill improvement by using SEAM is due to the more frequent model initializations and the more recent NWP model version of ENS extended. The paper is well written, the methodology and results are nicely presented and compared. The real value of this study is the application of products off the shelf (available operational products). Hence the results can be directly incorporated in real-time operational streamflow forecasting practice. The paper should be foreseen for publication in HESS after minor revisions.

Comments: p 2, l 24: Typo, replace TSYS4 with SYS4

p 2, l 30: please add the forecast length published in the seasonal outlook of EFAS

p 2, l 119: please add possible drawbacks of selecting a random member of SYS4 (one point was raised p 6 l 192- p7 l 195). Another possible drawback could be that ensemble members are combined originating from complete different climatological conditions day 1 – day 46.

p 3, l 89: Are the 5kmx5km grid cells of Lisflood further subdivided in elevation zones?

p 4, l 124: Are bias/drift correction methods applied to correct the meteorological forecasts?

p 5, l 135: the description of the hindcast period used in this study is a little bit confusing due to the mixture of forecast dates (2015-05-14 – 2016-06-02) used to produce the hindcast dataset and the forecast dates of the retrospective forecasts. Please clarify! One possibility would be probably to add the range of forecast dates. Something like: "...the hindcast data set of SEAM covers the period 1995-05-14 to 2016-06-02..." "...the SYS4 re-forecasts used in this study are initialized each month over the period 1995-05-01 to 2016-06-01..."

p 5, l 160: replace SEAS with SYS4

p 6, l 161: Incomplete sentence, I assume: ".... as in SEAM to account for the difference in ensemble size...."

p 7, l 206: Another option of the poor performance of Lisflood in these regions could be the snow modelling component. In steep orography a 5km x 5km grid is relatively coarse to model snow adequately, are grid cells of Lisflood further subdivided in eleva-

tion zones? Please add a comment/discussion of the snow modelling performance of Lisflood.

p 8, l 231: add Figure to the figure number "...Cologne (Figure 4)...".

p 8, l 233: I assume 3% of its climatological value is derived from the simulated climatology and not from the observed climatology? Please specify!

p 8, l 240: It should be mentioned that the second low flow event was hit by the SYS4 forecast initialized 2003-09-01. This signal towards a low flow event is missing in the SEAM forecasts published after 2003-09-01. In SEAM a signal towards an extreme low flow event first appears about 3 days before the begin of the event (forecast date 2015-09-14). I would add the real forecast dates to Figure 4 and not the forecast dates the hindcast data set is produced. This could be a little bit confusing for a reader not familiar to the hindcast procedure of ENS extended.

p 8 Conclusion: I miss a discussion of potential improvements of the presented seamless forecasting system. Are there any ideas how to reduce the higher spread of the CRPSS of SEAM compared to SYS 4 in figure 2 c, d? Probably an improvement of the methodology of the concatenation of the forecasts from the two systems? Please add this aspect to the conclusions. Another aspect I miss is the conclusion from Figure 2 b): The improved boundary condition of the first 46 days originating from the more recent model version with a higher resolution doesn't improve the predictability (forecast skill) after day 46.

Figure 3: Are all forecast dates used in this analysis? Please add to the caption to be consistent with the caption of Figure 2.
* * *

---

## Author Comment (AC1) · 24 Oct 2017

**Response to reviewer 1**

*Reviewers comment's in blue,* our responses in black

*In this paper the authors evaluate the added benefit of using a seamless integration (SEAM) of the outputs from ECMWF extended-range ensemble prediction system ENS-ER) and the ECMWF system 4 seasonal forecast system (SYS4) for hydrological applications. The added benefit from this approach is evaluated by comparing the continuous rank probability scores for the outputs from the hydrological model LISFLOOD forced by SYS4, SEAM, and a climatological ensemble (CLIM) over the hindcast period.*

*The authors find that hydrological hindcasts made using SEAM show better skill, over those made using SYS4, for much of Europe with lead times up to seven weeks. In some areas like the parts of Alps and northern Finland the reverse was true; however these results are uncertain due to the general poor performance of LISFLOOD in these regions. They argue that the increased skill can be attributed to the better initial conditions of the hydrological and meteorological conditions (models are initialised biweekly as opposed to once per month for SYS4) as well as the use of a better atmospheric model in SEAM (the atmospheric model used in SYS4 is locked at the initial version released with system 4 while the one used in ENS-ER is updated regularly). They conclude that the use of SEAM for hydrological forecasting at the seasonal scale has an added value for decision makers given the higher frequency of updates and improved skill, especially at the sub seasonal scale, making the forecasts more actionable.*

*The topic of this paper is of great interest at the moment considering the increased focus on forecasting at the sub seasonal to seasonal scales in recent years. Although the concept is not new this paper is the first, that I am aware of, that makes an attempt to evaluate a system that utilises current 'off-the-shelf' operational products. The pa per is well written with a good structure and generally clearly formulated, the methods are scientifically sound, and the results are interesting. Additionally, the research presented in this paper is very relevant to the topic of this special issue. In my opinion, the manuscript has a lot of potential for publication in this HESS special issue. However the authors need to clarify some points and revise some statements so that the paper is more easily understood.*

*General comments:*

*1) I feel that it is not clear for what periods the study was performed, something which has a bearing on the quality of the results. The authors state that (P5, L132-L135)*

*"This study focuses on the performance of SYS4 and SEAM over the hindcast period of the operational forecast with a sequence of starting dates over the period 2015-05-14 (the first available date with 11-member hindcast for ENS-ER) to 2016-06-02 producing daily output time series of discharge over the 20-year hindcast period."*

*The first part of the sentence suggests that the evaluation period is between the dates 2015-05-14 and 2016-06-02 yet the second part says that the hindcast period has a length of 20 years. The next line has a similar mixed message. From the paper I get the general impression that the evaluation is done for the 20 year period so I assume that the issue is to do with how section 2.3 is worded. This should be addressed as there is some confusion in the way that the paragraph (p5, L231-L238) explains it. Further it has implications on the robustness of the results, should the evaluation period be just the 13 months between the aforementioned dates this would give a limited data sample from which to draw the wider reaching conclusions made by the authors. How can the authors know*

*whether the performance of the different approaches during that period was typical of their general performance?*

The hindcast has 20 years of rerun forecasts, so it is not just one year of integration. The section has now been clarified and we have changed the wording to:

"This study focuses on the performance of SYS4 and SEAM over the hindcast period of the operational forecast. The hindcast of the ensemble forecast is produced twice per week (Mondays and Thursdays) by running an ensemble of 11 members with for that particular day and month, for each of the previous 20 years. The hindcast is run up to 46 days, similar to the ENS-ER. For this experiment, the hindcasts with a sequence of starting dates from 2015-05-14 (the first available date with 11-member hindcast for ENS-ER) to 2016-06-02 were used. This provided 13 monthly starting dates for SYS4 and 111 biweekly starting dates for SEAM with corresponding hindcast set covering all seasons over the previous 20-year period, each with 11 ensemble members. The output was averaged to weekly means before the skill score analysis."

Further, we will add a figure to explain how the hindcasts of the extended-range forecasts are set up.

*2) The results show that SEAM has skill over SYS4 in the first 3-8 weeks (Figure2b), mostly concentrated in the first 6 weeks. This would imply that there may be a benefit of merging the two meteorological forecasts before day 46. Did the authors consider this and if not why?*

We are not sure if we understand this comment. Fig2b shows that SEAM has more skill than SYS4 for the first 2-3 weeks, but that after that there are some areas where the SYS4 performs better and vice versa. The differences can have many explanations, where geography and altitude plays a part (Fig 3). For those areas where the SYS4 performs better than SEAM, it could as Kean suggests be beneficial to switch to SYS4 earlier than after 46 days. However, that would be interesting from an operational point-of view and is out of scope for this study. A system where you would switch between two systems in an optimal way would have to be carefully calibrated and the effect of switching forecasts would have to be significantly better to justify it. We would rather advocate that SYS4 is used in areas where it is clearly better than SEAM, or as a complement to SEAM. However, we will in future studies dive deeper into the differences in skill between the different forecasts.

*Specific comments:*

*P2, L26: "TSYS4 is also ..." I assume that this is a typo and should read, "SYS4 is also..."*

Yes, it was corrected to SYS4.

*P7, L224-L225: Although this line is factually correct it appears to contradict the preceding ones. The reader is being told how the low flows during this period caused substantial economic losses due to it affecting inland navigation in the Danube and Rhine basins only then to be told that navigations are regulated during high flows and not low flows. I suggest rewording this or removing this sentence to remove the perceived contradiction or removing this line altogether as it does not add anything significant to the discussion.*

The sentence was there to point to the fact that there are no regulated restrictions on the low flow; it is down to the transport companies to make the decision. We agree that it does not add any significant information and the sentence will be deleted in the revised version..

*P8, L249-L250: The second part of this line is awkward to read and should be rephrased.*

The sentence was rephrased to: "The onset of the second low period was correctly modeled by the SEAM system, whereas the timing of the low flow was missed by SYS4"

*P11, L350-L35: I think the reference is - Pappenberger, F., Wetterhall, F., Dutra, E., Di Giuseppe, F., Bogner, K., Alfieri, L., and Cloke, H. L.: Seamless forecasting of extreme events on a global scale, pp. 3–10, Proceedings of H01, IAHS-IAPSO-IASPEI Assembly, Gothenburg, Sweden, July 2013 (IAHS Publ. 359, 2013)*

Yes, that is correct; the reference has now been updated.

*P16, Caption to figure 3: The last line states, "The dimension of the circles is proportional to the number of days while the color scale refers to progressive weeks." What do the authors mean by number of days?*

The size of the circles are proportional to the number of day of predictability. The circle size was missing in the plot legend that has now been revised. To make the plot more readable we had also colour-coded the circles in broad weekly changes. Clearly, there is a correlation between colour and circle sizes as the darker the colour the larger the symbol dimension. However we found that the colour breaks made the plot more readable. The graphics of the plot has been slightly revised and is as follows:

[Figure]

---

## Author Comment (AC2) · 24 Oct 2017

*B. Klein*

*The manuscript shows the development and the skill of a seamless hydrological forecasting system from sub-seasonal to seasonal scales. Meteorological forecasts from ENS extended (day 1 – 46) and SYS4 (47 to month 7) are merged by randomly selecting ensemble members of SYS4 after ENS extended ends. The skill analysis shows that most of the skill improvement by using SEAM is due to the more frequent model initializations and the more recent NWP model version of ENS extended. The paper is well written, the methodology and results are nicely presented and compared. The real value of this study is the application of products off the shelf (available operational products). Hence the results can be directly incorporated in real-time operational streamflow forecasting practice. The paper should be foreseen for publication in HESS after minor revisions.*

Thank you very much Bastian for the comments. Below are detailed point-to-point answers to your remarks, but we have also taken on board your comments of expanding the discussion and conclusion part on the predictability and limitations of the two systems and will expand the discussion part further.

*Comments:*

*p 2, l 24: Typo, replace TSYS4 with SYS4*

The typo has been corrected.

*p 2, l 30: please add the forecast length published in the seasonal outlook of EFAS*

"...with a lead-time of 7 month." was added to clarify.

*p 2, l 119:  please add possible drawbacks of selecting a random member of SYS4 (one point was raised p 6 l 192- p7 l 195).  Another possible drawback could be that ensemble  members  are combined  originating  from  complete  different  climatological  conditions day 1 – day 46.*

We are aware of this problem and we tried to address it on p6, but will expand on this and discuss the drawbacks further. However, the regimes over Europe can shift quite rapidly and it is not certain that matching the ensembles would increase the skill of the seamless.

*p 3, l 89: Are the 5kmx5km grid cells of Lisflood further subdivided in elevation zones?*

Yes, they are divided into three sub elevation zones to account for differences in snow accumulation and snowmelt. See more details in the answer to P7 below.

*p 4, l 124: Are bias/drift correction methods applied to correct the meteorological forecasts?*

No bias correction is applied to the meteorological forecasts

*p 5, l 135: the description of the hindcast period used in this study is a little bit confusing due to the mixture of forecast dates (2015-05-14 – 2016-06-02) used to produce the hindcast dataset and the forecast dates of the retrospective forecasts.  Please clarify!*

*One possibility would be probably to add the range of forecast dates. Something like: "...the hindcast data set of SEAM covers the period 1995-05-14 to 2016-06-02..." "...the SYS4 re-forecasts used in this study are initialized each month over the period 1995-05-01 to 2016-06-01..."*

Thank you for the suggestion. Also Kean commented on the difficulty of understand the setup of the experiment. We have taken care in explaining the hindcast and experiment setup more in detail. We will also add a figure to explain the setup of the hindcast system.

Corrected.

*p 6, l 161: Incomplete sentence, I assume: ".... as in SEAM to account for the difference in ensemble size...."*

The incomplete sentence was deleted

*p 7, l 206: Another option of the poor performance of Lisflood in these regions could be the snow modelling component. In steep orography a 5km x 5km grid is relatively coarse to model snow adequately, are grid cells of Lisflood further subdivided in elevation zones? Please add a comment/discussion of the snow modelling performance of Lisflood.*

The snow modelling in LISFLOOD is a degree-day method with elevation zones to further differentiate the snow processes in steep orography. This could explain differences in the model performances if the results were compared with observed runoff. However, the model results are compared with a climatology run using observed precip and temperature, and it is more likely that the poor NWP representation of temperature and precipitation are the culprits.However, the snow modelling component could also play a role in this, and we will add a description and discussion on this to the paper.

"The snow accumulation and snowmelt are further divided into three elevation zones within a grip in LISFLOOD to better account for orographic effects in mountainous regions. However, this increase in sub grid resolution is not likely to be high enough to capture the snow variability during the snow accumulation and snowmelt in mountainous regions. Further, precipitation forecasts have documented biases in steep orography (Haiden et al., 2014).

*p 8, l 231: add Figure to the figure number "...Cologne (Figure 4)...".*

Corrected

*p 8, l 233: I assume 3% of its climatological value is derived from the simulated climatology and not from the observed climatology? Please specify!*

Yes, it is correct, we are throughout the paper comparing against modelled climatology. We have taken care to make this very clear wherever this is mentioned in the paper. At the above mentioned passage we have changed the sentence to: "went below the 3% percentile of *the modelled* climatological value". Italics denote the addition

*p 8, l 240: It should be mentioned that the second low flow event was hit by the SYS4 forecast initialized 2003-09-01. This signal towards a low flow event is missing in the SEAM forecasts published after 2003-09-01. In SEAM a signal towards an extreme low flow event first appears about 3 days before the begin of the event (forecast date 2015-09-14). I would add the real forecast dates to*

*Figure 4 and not the forecast dates the hindcast data set is produced. This could be a little bit confusing for a reader not familiar to the hindcast procedure of ENS extended.*

Yes, and the example is chosen to illustrate a situation where the SYS4 performed well. We also point to the fact that SYS4 does perform well in this particular case in the discussion. However, the higher frequency of the SEAM would give it an advantage when you are closer to the event, since you would get more detailed information about the timing. The following was added to stress the point: "SYS4 does indicate the second low flow with a longer lead time than SEAM. However, SYS4 misses the timing of the event.

Figure 4 was also improved to show more clearly the forecast dates vs the verification dates.
*p 8 Conclusion: I miss a discussion of potential improvements of the presented seamless forecasting system. Are there any ideas how to reduce the higher spread of the CRPSS of SEAM compared to SYS 4 in figure 2 c, d? Probably an improvement of the methodology of the concatenation of the forecasts from the two systems? Please add this aspect to the conclusions.*

This is a good point, and still be investigated, however outside the scope of this paper. We will add the following to the Conclusions.

"Future work with the seamless forecasting system is to further explore the limits of predictability to assess the strengths and limitations of the current setup. The assumption that the forecasts can be randomly concatenated would also need to be tested against a system where the forecasts are matched according to their respective climatology."

*Another aspect I miss is the conclusion from Figure 2 b):*
*The improved boundary condition of the first 46 days originating from the more recent model version with a higher resolution doesn't improve the predictability (forecast skill) after day 46.*

This is also a good point, and more is to say on the predictability of the two forecast systems. This will however as mentioned above be dealt with in another study, so we are reluctant to speculate too much at this time.

*Figure 3: Are all forecast dates used in this analysis? Please add to the caption to be consistent with the caption of Figure 2.*

No, in Figure 3 only the first forecast of the month is used to avoid too much the effect of the initial conditions of the hydrological model. This will be clarified in the revised manuscript.

---

## Short Comment (SC1) · 30 Oct 2017

This paper has a good (not new) idea, but is disappointing as it just skims over results without proper analysis. It currently does not have a proper scientific discussion and reads like it was rushed. In addition, the paper seems to have written for a different journal, it is extremely short (which is good in theory), but simply lacks depth and proper analysis. Results are not properly explained and leave many questions. This is best illustrated by the use of a single score, which does only measure one property of an ensemble forecast - I would have at least expected some de-compositions.

Detailed comments: Acronym ENS-ER appears in introduction first and needs to be

defined in introduction not only abstract. I could not find that acronym on ECWMF's websites which makes me wonder what the authors have actually used.

The introduction defeats most of the paper. I clearly states: "This implies that the skill of SYS4 is lower relative to ENS-ER in the overlapping first six weeks (Di Giuseppe et al., 2013)", which is obviously a result that has been already published by one of the authors earlier

L34 it is unclear why the extension leads to benefit. Point (ii) - that has been possible before, what is better and why? There are no references stated for the hypothesis listed in (i) to (iii) - a more detailed in depth discussion and reasoning (or supporting results) are needed

"The extended lead time provided by running EFAS forced by weather prediction across different time scales could potentially provide added benefit in terms of very early planning, for example for agriculture, energy and transport sectors as well as water resources management." - where is the evidence for that statement? references? Studies - this unsubstantiated and symptomatic for the rest of the paper - many claims or statements which are not backed up.

"often model implementation is segmented for practical reasons. Still major efforts have been made to create unified systems" - it is completely unclear what is meant - clarify

"Similarly, the UK Met Office has in the past twenty-five years worked to create a unified model that could work across all scales (Brown et al., 2012). Also the climate community has moved in the same direction. For example, the EC-Earth project shows that a bridge can be made between weather, seasonal forecasting and beyond (Hazeleger et al., 2010, 2012)." this is not relevant for the paper. I am unsure what point the authors are trying to make with respect to the hypothesis tested in this paper. Introduction needs significant shortening.

"avoiding the complications of new developments while generating forecast products

to meet different types of users (Pappenberger et al., 2013)." Pappenberger is clearly wrong - one will always need different products for different applications.

"diverge over time, only re-converging when the seasonal system" That assumes that the seasonal system is very close to the system from which it is derived from. I just googled ECMWF System 5 and it seems to come from an older model cycle, hence this statement is clearly incorrect

"final products should be provided in terms of anomalies calculated against the model climate" that assumes that the model universe behaves similarly to the real universe in terms of anomalies - can the authors provide any prove and evidence?

"What is the gain of using a more recent model version in the first 46 days provided by the use of the ENS-ER?" I don't understand that question cause according to the authors this has been already answered in a paper cited by the authors themselves, (Di Giuseppe et al., 2013). It demonstrate that the paper currently only presents a very very incremental step.

It is unclear how the authors come to 786 reference points - how have they been choosen - the claims made by the authors are not substantiated by the results presented. Can the authors please add the analysis which lead to those points? this is a clear example where the paper has been cutting corners rather than explaining properly what has been done.

" (referred to as tuning in the NWP nomenclature)" This is a hydrology journal, why do you explain that?

"Using the WB run as proxy observation simplifies the interpretation of the skill scores as it avoids the complication of having to assess the bias against observed discharge." This maybe convenient to do, but then the analysis could have been done against all grid points or far more ($\sim$700 is pretty low given the size of that Grid). The authors need to elaborate on the limitations this analysis places on the results of the study. I

am also thoroughly confused, the authors said that they had real observations for the calibration. I would expect at least some analysis against those real observations. Far more detail needs to be provided.

"The hindcast period can together with observations be employed to calibrate the forecast in an operational setting (Di Giuseppe et al., 2013)." I am unsure about what the authors mean with that statement and find the reference strange and forced (deliberate self citation?). Can the authors please cite references from others too?

Figure 1 is unclear - how do different ensemble number play a role. Did you only merge 11?

2.3. Experimental set-up - you are comparing apples with pears. One system has clearly a much larger sample size and the authors do not explain how the adjust for that fact. Results cannot be robust unless this is taken into account. Please revise your method thoroughly.

CRPS is equalised by randomly drawing from the distributions - that is at odds with the statistical literature. Check for example this presentation: http://empslocal.ex.ac.uk/people/staff/ferro/Presentations/ems2013ferro-fair.pdf

The authors need to present more scores and analysis. They talk explicitly about droughts in the introduction - this scores does not analyse. To understand skill, one needs to look at least at the decompositions of the CRPS. The analysis needs to be extended significantly and far better discussed.

"then some points show a benefit of using the SYS4 instead of SEAM." - why? explain

"In the above example, a decision maker would have to make a decision based on a forecast that was issued 2.5 weeks earlier, which would inherently make the decision more uncertain if you only had the seasonal forecast. With the seamless system available a decision maker would gain the same early indication of a hazardous event and also have the benefit of frequent updates." Can the authors please test their hypothesis

and provide prove for such unsubstantiated statements? where is the social scientific evidence?

I do not understand the point of section 3.3. - it presents a single case and then makes some wild statements. Please assemble a larger number of cases or simply cut.

the analysis overall falls short for more details. It simply skims over results without really going into them and properly analysing them. Many hydro aspects are ignore.

Please explain how your results are driven by spatial variations of the weather forecasts.

Conclusions are not comprehensive enough and a proper scientific discussion is missing.

---

## Author Comment (AC3) · 1 Nov 2017

**Mike Hardeker**

*This paper has a good (not new) idea, but is disappointing as it just skims over results without proper analysis. It currently does not have a proper scientific discussion and reads like it was rushed. In addition, the paper seems to have written for a different journal, it is extremely short (which is good in theory), but simply lacks depth and proper analysis. Results are not properly explained and leave many questions. This is best illustrated by the use of a single score, which does only measure one property of an ensemble forecast - I would have at least expected some de-compositions.*

Thanks for your comments. This paper is purposely short as we intended to showcase only one result: how much is the gain in using a seamless forecast system instead of the seasonal forecast. As you also point out this idea is not new and has been referred to in many other papers. However, to our knowledge, this is the first paper that quantifies what is the effective gain in an operational system in weeks of predictability. As such, this paper tries to diagnose the advantage of a concatenated system against the exclusive use of System-4 which is often the preferred choices.

Is it true that this paper leave questions open. The most urgent one in our view is what happens if someone has only access to the seasonal system? This is quite common as seasonal forecast is freely available as opposite to the ENS forecast. There are ways to improve the predictability of the seasonal forecast for example by applying the finding of this paper. We are preparing another work looking specifically to this aspect and exploring in details the sub-seasonal to seasonal predictability.

We want to keep this work as much as possible focused on this single question, however, we agree that other diagnostic could be added and we will extend the results including bias and reliability in the revised version, and also extend the discussion.

*Detailed comments: Acronym ENS-ER appears in introduction first and needs to be defined in introduction not only abstract. I could not find that acronym on ECWMF's websites which makes me wonder what the authors have actually used.*

We have added the explanation of the acronym to the text. The extended range ensemble prediction system (ENS-ER) refers to the bi-weekly 46 days extension of the otherwise daily ensemble prediction. ENS is the official acronym for the ensemble forecast, and we added ER to distinguish this from the normal ENS. Even if this is not an "official" ECMWF naming convention, we found this to be a useful acronym for this paper

*The introduction defeats most of the paper. I clearly states: "This implies that the skill of SYS4 is lower relative to ENS-ER in the overlapping first six weeks (Di Giuseppe et al., 2013)", which is obviously a result that has been already published by one of the authors earlier.*

The idea of the paper is not to assess the fact that the ENS-ER is a better forecast, it is how much better it is and whether it can be used in together with SYS4 to create a seamless forecast, which is updated more frequently than the seasonal forecast. This is important information for any user of hydrological seasonal forecasts, as was also pointed out by the other reviewers.

Also in (Di Giuseppe et al., 2013) we assumed that the ENS-ER in the overlapping first six weeks was better that than System-4 and then went on doing other analysis. The fact that this assumption has always been accepted without questioning reinforces the idea of this paper, which tries to quantify those statements. Di Giuseppe et al., 2013 looks at forecast calibration for the purposes of generating a malaria early warning system. There is almost no overlap with what done here apart from the use of the ENS long range forecast.

*L34 it is unclear why the extension leads to benefit. Point (ii) - that has been possible before, what is better and why? There are no references stated for the hypothesis listed in (i) to (iii) - a more detailed in depth discussion and reasoning (or supporting results) are needed.*

Regarding point 2:, the previous hindcast of the monthly extension was only 5 member and up to day 32. The new system with 11 members and lead-time 46 days is much more useful than the previous system, therefore the possibilities of carrying out pre-and post-processing has greatly increased.

The first point is related to the fact that we can now extend the ensemble forecast more in time than was possible previously, and that we can issue seasonal forecasts with higher frequency than before, given that the method of concatenation ito to a seamless forecast is working well.

The third point is a bit more speculative, but a decision support system for more products that was previously available would be possible and feasible to implement. The examples of benefits are given in the below statement.

We will expand on these three points and support them with references.

*"The extended lead time provided by running EFAS forced by weather prediction across different time scales could potentially provide added benefit in terms of very early planning, for example for agriculture, energy and transport sectors as well as water resources management." - where is the evidence for that statement? references? Studies - this unsubstantiated and symptomatic for the rest of the paper - many claims or statements which are not backed up.*

The statement is quite modest. We are simply saying the availability of a skilful forecast X days ahead is more useful than a skilful forecast provided Y days ahead if X>Y. We would imagine this to be uncontroversial. If in some sectoral application there is only need for Y days forecast, then the X days information can be easily disregarded. Forecasts are used in many applications, and we will substantiate that with more references to such studies.

*"often model implementation is segmented for practical reasons. Still major efforts have been made to create unified systems" - it is completely unclear what is meant - clarify*

As the reviewer points out our introduction is quite long as we were very comprehensive in highlighting the context from which this paper was generated. We have explained that the various weather prediction systems have been developed from requirements that have been added in time as weather forecast has improved in terms of predictability. This has led to fragmented systems. This fragmentation is somehow not intentional, however practical. Some institutions have gone all the way to rewrite their model (UKMET office) so that this could be used at all time scales. These systems could provide possibly a better tool for predictability studies. However, this work does not try to quantify predictability per-se but to put a predictability length to one of the most used system in the world, given that one takes what is available from the shelf. If the reviewer is searching for a theoretical study this is not the right paper. As the two other reviewers have pointed out this paper has value as it analyses in the specific a very well used system even if the results validity are then limited to that particular system.

*"Similarly, the UK Met Office has in the past twenty-five years worked to create a unified model that could work across all scales (Brown et al., 2012). Also the climate community has moved in the same direction. For example, the EC-Earth project shows that a bridge can be made between weather, seasonal forecasting and beyond (Hazeleger et al., 2010, 2012)." this is not relevant for the paper. I am unsure what point the au-thors are trying to make with respect to the hypothesis tested in this paper.*

This sentence is instead quite relevant as it compares our concatenation approach to another approach (creating a unified model) that exists even if it is not used in this paper. We believe it is part of the bibliographic review process in the introduction to acknowledge what is available even if is not used.

*Introduction needs significant shortening.*

Sorry but we disagree as we find our introduction quite a nice historical overview of the conception, the designs and the different approaches followed for the practical implementations of seamless forecasts.

*"avoiding the complications of new developments while generating forecast products to meet different types of users (Pappenberger et al., 2013)." Pappenberger is clearly wrong - one will always need different products for different applications.*

We are not arguing that a specific users do not need to tailor the product to meet their needs, just that you can achieve quite a lot with already existing information. The two systems, the seasonal and the extended range are both worth using to a larger extent than they currently are, and they are readily available. The tailoring towards your own needs is necessary for any application as you clearly state, and that is exactly what we are doing when we are using the meteorological forecast to force a hydrological forecast.

*"diverge over time, only re-converging when the seasonal system" That assumes that the seasonal system is very close to the system from which it is derived from. I just googled ECMWF System 5 and it seems to come from an older model cycle, hence this statement is clearly incorrect*

We agree, the statement is too strong, the systems never completely converge, the gap in model cycles are shortened with a new release of a seasonal forecast. We have changed the wording to:

"One important consequence of this difference in design is that, for example, the much more frequent updates to the extended range compared to the seasonal system at ECMWF, imply that the bias characteristics of the two systems diverge over time, only closing when the seasonal system is updated."

*"final products should be provided in terms of anomalies calculated against the model climate" that assumes that the model universe behaves similarly to the real universe in terms of anomalies - can the authors provide any prove and evidence?*

Yes, the EFAS system behaves well in terms of issuing forecasts in comparison with the model climatology. It is not perfect, so is no system. EFAS has been calibrated against observations where they are available, and the performance is generally good. The praxis of EFAS is to compare against its water balance, this is the standard procedure. We can be clearer in the references to previous studies regarding this.

This argument here is rather that the concatenation itself needs to be taken care of since it is likely to create a bias when the two systems are combined. We have added a sentence to point tos this argument.

*"What is the gain of using a more recent model version in the first 46 days provided by the use of the ENS-ER?" I don't understand that question cause according to the authors this has been already answered in a paper cited by the authors themselves, (Di Giuseppe et al., 2013). It demonstrate that the paper currently only presents a very very incremental step.*

In Di Giuseppe et al 2013 we assumed that a seamless system would have been better than the seasonal forecast, however we never proved it neither we looked at the differences with system-4.In this paper we are actually proving what is the benefit of using a seamless system.

*It is unclear how the authors come to 786 reference points - how have they been choosen - the claims made by the authors are not substantiated by the results presented. Can the authors please add the analysis which lead to those points? this is a clear example where the paper has been cutting corners rather than explaining properly what has been done.*

This will be more clearly explained. We also apologise for an error, the final number of reference points were 679, not 786 as originally stated. The reference are the EFAS outlets from the several sub catchments in the domain. They were chosen as representative points for the performance, and

were the points that were used for the operational calibration of EFAS. We will state this more clearly in the paper. We will also add references to the literature where more detail can be found.

However, we do not understand the comment on why the claims are not substantiated by the results? In fact, we could have choses a random number of points, or all of them, and the results would still have been valid as long as we are comparing against a modelled water balance. The selection of these particular ones was to have a reasonable number of points with a good geographical spread to assess the performance of the system.

*" (referred to as tuning in the NWP nomenclature)" This is a hydrology journal, why do you explain that?*

The journal is read by both meteorologists and hydrologists. Often the two communities use different nomenclature for the same process. We do not think there is any harm to explicitly clarify this aspect for the benefit of a vaster reader audience.

*"Using the WB run as proxy observation simplifies the interpretation of the skill scores as it avoids the complication of having to assess the bias against observed discharge." This maybe convenient to do, but then the analysis could have been done against all grid points or far more (*
*700 is pretty low given the size of that Grid). The authors need to elaborate on the limitations this analysis places on the results of the study. I am also thoroughly confused, the authors said that they had real observations for the calibration. I would expect at least some analysis against those real observations. Far more detail needs to be provided.*

To answer the first comment on the number of points used for the assessment of the system. The total number of points at which discharge is calculated over all of EFAS is 38452. We could have calculated the performance on each of these points, and we routinely do that as part of our performance. However, since they would in many cases be highly correlated (points along the same river will behave similar), a sub-sampling was made to represent the performance over the entire domain. This was a conscious decision to simplify the calculations and to avoid too correlated skill scores, as independent sampling as possible. We consider the selection good enough to represent the performance of the system and do not see the reason to increase the number of points.

The second question regarding why we did not include the observational data has been discussed in the paper. The EFAS system is covering the entire European continent and can as such not be perfectly calibrated everywhere, especially not on a 5km grid. The observations are alos not available for the full hindcast period at each location.

The water balance run, which is the model performance using observed precipitation and temperature, are a proxy for observations, and is what we chose to compare the performance of the models against. The benefits of using the water balance is to avoid observational errors and also to mimic the performance of the operational EFAS forecast system, where the forecasts are also compared with the water balance rather than observations. Since we are comparing the two forecasting systems and not trying to assess the total skill of EFAS, the use of the water balance run is justified. We understand that this was not fully explained in the paper, and will add this to the discussion.

*"The hindcast period can together with observations be employed to calibrate the forecast in an operational setting (Di Giuseppe et al., 2013)." I am unsure about what the authors mean with that statement and find the reference strange and forced (deliberate self citation?). Can the authors please cite references from others too?*

This paper is cited as an example of a correction that can be calculated from the hindcast set and then applied to the forecast. The methodology developed in Di Giuseppe et al 2013 is quite complex as it was designed to correct a precipitation systematic southerly shift in the west African monsoon. However, the calibration was implemented for the exact same system used here, i.e. a seamless concatenation of the ENS-ER and system-4. For this reason, we thought it was a well suited

reference. However, following the suggestion we have added other two well-known work for bias correction.

*Figure 1 is unclear - how do different ensemble number play a role. Did you only merge 11?*

We have added a new schematic, which explains in better details how the hindcast set from the seamless, is constructed. Since there are only 11 hindcasts of ENS-ER, only 11 could be merged with the hindcast of the seasonal forecast.

*2.3. Experimental set-up - you are comparing apples with pears. One system has clearly a much larger sample size and the authors do not explain how the adjust for that fact. Results cannot be robust unless this is taken into account. Please revise your method thoroughly.*

This is taken into account in the analysis (see figure 2) where we compare only the hindcast from the first of the month from SEAM with the seasonal forecasts, therefore not using all forecasts from SEAM. Same as in figure 3, where we only use the first forecast of the month from SEAM in comparison with SEAS. We thought it useful to show the performance over the entire period in figure 2a, therefore it was added. We will make the this clearer in the description of the methodology..

*CRPS is equalised by randomly drawing from the distributions - that is at odds with the statistical literature. Check for example this presentation: http://empslocal.ex.ac.uk/people/staff/ferro/Presentations/ems2013ferro-fair.pdf*

The random drawing of members from the SYS 4 distribution would not induce a large error, since the members are interchangeable. However, this will be corrected in the revised version where we will instead use the 15-members ensemble from the SYS4 and then corrected for the variation in ensemble size, as suggested by Ferro et al, 2008.

*The authors need to present more scores and analysis. They talk explicitly about droughts in the introduction - this scores does not analyse. To understand skill, one needs to look at least at the decompositions of the CRPS. The analysis needs to be extended significantly and far better discussed. "then some points show a benefit of using the SYS4 instead of SEAM." - why? explain*

We mention droughts and low flows as possibly uses for a seasonal forecasting system; we do not state that we will look into to it in this paper. We will add reliability and bias to the analysis, as stated earlier. The better performance of SYS4 at certain locations is not strange, we do not expect SYS4 to be outperformed at all locations. However, the exact reasons for the better performance for each location is beyond the scope of this paper and will be more looked at in later studies.

*"In the above example, a decision maker would have to make a decision based on a forecast that was issued 2.5 weeks earlier, which would inherently make the decision more uncertain if you only had the seasonal forecast. With the seamless system available a decision maker would gain the same early indication of a hazardous event and also have the benefit of frequent updates." Can the authors please test their hypothesis and provide prove for such unsubstantiated statements? where is the social scientific evidence?*

In this statement we are just are just stating that a decision substantiated by the availability of more detailed information is more robust and less of guesswork. We refrain from any speculations as to what the implications are for the human intervention in the forecast process, merely that the forecast is substantially better and more importantly, more frequently updated.
The only situation we can foresee in which this statement could be misleading is if the seamless forecast were not as accurate as the seasonal forecast, in which case a bad information might be worse than no information at all. As this is not the case and, it has been clearly proven throughout the paper, we do not see how this statement can be considered "unsubstantiated" as it is just driven by common sense.

*I do not understand the point of section 3.3. - it presents a single case and then makes some wild statements. Please assemble a larger number of cases or simply cut*

Section 3.3 does not claim to provide any statistical significance of the quality of SEAM against SYS4. This is done in the sections before. Here we have made a practical example for a case studies looking at what the more timely information provided by SEAM could imply in a decision making context. We believe the discussion that follow from figure 4 is not "wild" instead tries to explains in, admittedly, a simplified scenarios, which kind of product improvements could be achieved given the availability of the seamless system.

*the analysis overall falls short for more details. It simply skims over results without really going into them and properly analysing them. Many hydro aspects are ignore. Please explain how your results are driven by spatial variations of the weather forecasts.*

We understand that the suggestion is to perform a full sensitivity study of the presented results looking at the predictability arising from weather regimes /patterns. Looking at the hydrological predictability at different time scales as driven by weather is certainly an extremely worth matter, however it would require an analysis that is outside the scope of this paper and we see this more like the subject of upcoming studies.

*Conclusions are not comprehensive enough and a proper scientific discussion is missing*

Section 3 is named "result and discussion". As a matter of style preference we have decided to detailed discuss our results in this part of the paper. In the conclusions we only highlight the main novel aspects of the paper without repeating the discussion which takes place in session 3.
As the paper aims at answering one

**References**

Di Giuseppe, F., Molteni, F., and Tompkins, A. M. (2013) A rainfall calibration methodology for impacts modelling based on spatial mapping, Quarterly Journal of the Royal Meteorological Society, 139, 1389–1401.

Ferro, C. A. T., Richardson, D. S. and Weigel, A. P. (2008), On the effect of ensemble size on the discrete and continuous ranked probability scores. Met. Apps, 15: 19–24. doi:10.1002/met.45

---

## Referee Report (RR1)

REVIEW of the paper

**The benefit of seamless forecasts for hydrological predictions over Europe**

Authors: Fredrik Wetterhall and Francesca Di Giuseppe
Manuscript Number: hess-2017-527
Submitted: Hydrol. Earth Syst. Sci. Discuss.

This paper evaluates the performance of the hydrological forecast by merging sub-seasonal and seasonal rainfall forecasts. As expected authors found that the hydrological forecasts with merged rainfall forecasts are better than with the seasonal rainfall forecasts for first few weeks. Given that this is revised submission, I had opportunity to read the responses from the authors to address the comments given by the referees from initial submission. The authors have addressed most of those comments and revised the paper accordingly. However there are still some minor issues and errors which needs to be addressed.

The authors have compared hydrological forecasts forcing by merged rainfall forecasts with the seasonal forecasts (SYS4). I wonder whether they have also compared merged rainfall forecast with the SYS4?

Page 3, Line 60-65: These sentences are misleading and confusing. How simple concatenation of the best forecast can be complex in the simplification, then the concatenation is technically difficult?

P 3 Line 69: Can authors explain what the bias characteristics are?

Page 3, Line 91: delete "such as"

Figure 1 is not needed. It does not show anything more than the text in caption.

Page 5, Line 130: I suggest authors to explain in one sentence or two "mass-conserving interpolation".

Page 5, Line 137-138: "full hydro-meteorological integrations", but never see this term for the rest of the paper.

Page 5, Line 140-145: What is the actual period of forecast evaluation? Is it 2015-05-14 to 2016-06-02 or 1995 to 2015. I remembered this was raised by referees in earlier submission, but still it is not clear in the paper.

Page 5, line 147: Authors mentioned ENS-ER issued Monday and Thursday not Wednesday?

Page 5, Line 161: "modeled discharge", did not authors define this as WB runs. I suggest to use consistent terminology throughout the paper.

Page 6, Line 168: replace "*tof*N" to "*n, N*"

Page 6, Line 169: Define "RPS"

Page 6, Line 185: By definition forecast error is difference between observation and forecast, so need to say "against observations". It is better to give formula for mean relative error as some people define error as forecast – observation.

Figure 3: Legends for black solid lines should be median seasonal (not 10/90 percentiles) in c) and d). I can assume that 10 and 90 percentiles are computed from CRPSS of all river gauges, but never mentioned in the paper.

Page 7, Line 193: In figure 3a, the CRPSS at week six is less than 0.1 (not 0.2).

Page 7, Line 193: All river point? Do authors mean all river gauges?

Figure 3a: I think it is not fair comparison between SYS4 and SEAM when all start dates in SEAM are considered. Firstly, the sample sizes of both forecasts are not same, then more importantly the target dates of given lead time are not same.

Page 7, Line 215-219: Not sure what the authors want to say "An explanation can  …. driving forecast is used."

Figure 4 is hard to understand. "The dimension of the circles is proportional to the number of days while the color scale refers to progressive weeks". Number of days of what?

Page 8, Line 228: effect?

Page 8, Line 236-240: Please rephrase this paragraph. What do authors mean by recent "development of the precipitation forecast?"

Figure 5 caption: replace "functionality" with "function"

Figure 5: Legend for SYS4 is wrong (see previous comments)

Figure 7: In the previous figures, authors have shown 10 and 90 percentile, why they chose 25 and 75 percentiles. How are the percentiles computed for reliability diagrams?

Page 8, Line 252: "Both forecast systems are over-confident". How? For forecast probability less than 0.5, it is lower than the observed frequency, while for higher than 0.5, it is higher than the observed frequency.

---

## Author Response (AR2)

Response to reviewer 3

This paper evaluates the performance of the hydrological forecast by merging sub-seasonal and seasonal rainfall forecasts. As expected authors found that the hydrological forecasts with merged rainfall forecasts are better than with the seasonal rainfall forecasts for first few weeks. Given that this is revised submission, I had opportunity to read the responses from the authors to address the comments given by the referees from initial submission. The authors have addressed most of those comments and revised the paper accordingly. However there are still some minor issues and errors which needs to be addressed.

The authors have compared hydrological forecasts forcing by merged rainfall forecasts with the seasonal forecasts (SYS4). I wonder whether they have also compared merged rainfall forecast with the SYS4?

Response: No, we have not looked at the rainfall in particular. The idea of this paper was to focus on the discharge, as it is an integrating variable. It is not possible to disentangle the sources of bias without specifically looking at all elements of the water balance, and this will be done as the next step.

Page 3, Line 60-65: These sentences are misleading and confusing. How simple concatenation of the best forecast can be complex in the simplification, then the concatenation is technically difficult?

R: The term simple is meant to be referring to the fact that you do not have to create a new model that produces daily forecast from day 1 up to 6 months for each day of the year. We have changed the sentences to make it clearer:

*"The seamless idea could be translated into a concatenation of "the best" forecast at each lead-time. The clear advantage of this off-the-shelf seamless prediction conversion is that it utilizes products that are already available and operational, thereby avoiding the complications of new developments, while at the same time generating forecast products to meet different types of users (Pappenberger et al., 2013). There is however an underlying complexity in this simplification; the difference in design between the various forecasting systems makes the concatenation not entirely straight-forward. The forecasting systems are related since they are from different generations of the same model development, however they have non-matching temporal and spatial resolutions, different hindcast span and different ensemble sizes."*

P 3 Line 69: Can authors explain what the bias characteristics are?

R: The bias characteristics means simply the model error. We have changed the text to this:

*"One important consequence of this is that the more frequent updates to the extended range compared to the seasonal forecasting system at ECMWF causes the model errors from the two systems to diverge over time, and only closing this gap when the seasonal system is updated to a newer model version.."*

Page 3, Line 91: delete "such as"

R: This would change the meaning of the sentence, instead the whole section "such as soil and ground water interactions" was deleted.

Figure 1 is not needed. It does not show anything more than the text in caption.

R: The figure was added because it was requested to explain the hindcasts better. We think that this figure helps explains how the reforecasts are created and that it is therefore justified.

Page 5, Line 130: I suggest authors to explain in one sentence or two "mass-conserving interpolation".

*R: This was added: "The mass-conservative interpolation summarizes the partial contribution of the meteorological input fields onto the LISFLOOD grid."*

Page 5, Line 137-138: "full hydro-meteorological integrations", but never see this term for the rest of the paper.

R: The terms that are used throughout the paper are SEAM and SYS4, which refer to the hydrometeorological integrations rather than just the meteorological forcing. The sentence was changed to:

*For simplicity SYS4 and SEAM will from now on refer to the full hydro-meteorological model chain and not only the meteorological forcing for the remainder of this paper*

Page 5, Line 140-145: What is the actual period of forecast evaluation? Is it 2015-05-14 to 2016-06-02 or 1995 to 2015. I remembered this was raised by referees in earlier submission, but still it is not clear in the paper.

R: The period of evaluation are the hindcasts starting at 2015-05-14, (which are 1995-05-14 - 2014-05-14) to 2016-06-02, which have the corresponding hindcast 1996-06-02 - 2015-06-02.

This is very difficult to explain in words, there it was accompanied with Fig 1. I also added a sentence *"As described above, the hindcasts are the reforecasts over the previous 20 years and is produced for each individual run of the ENS-ER"*

Page 5, line 147: Authors mentioned ENS-ER issued Monday and Thursday not Wednesday?

R: Corrected to Thursday.

Page 5, Line 161: "modeled discharge", did not authors define this as WB runs. I suggest to use consistent terminology throughout the paper.

R: Thanks for that observations, we changed this to WB

Page 6, Line 168: replace "tofN" to "n, N"

R: Replaced with : "step *t* of *N* number…"

Page 6, Line 169: Define "RPS"

R: Corrected to "ranked probability score (RPS)"

Page 6, Line 185: By definition forecast error is difference between observation and forecast, so need to say "against observations". It is better to give formula for mean relative error as some people define error as forecast – observation.

R: Formula for MRE was added

Figure 3: Legends for black solid lines should be median seasonal (not 10/90 percentiles) in c) and d). I can assume that 10 and 90 percentiles are computed from CRPSS of all river gauges, but never mentioned in the paper.

R: The 10 and 90th percentiles are computed from the 679 outletpoints that were used for initial calibration. This was added to the figure text. Figure 3 was also corrected regarding the median and 10 and 09th percentiles.

Page 7, Line 193: In figure 3a, the CRPSS at week six is less than 0.1 (not 0.2).

R: Yes, this was corrected

Page 7, Line 193: All river point? Do authors mean all river gauges?

R: Was changed to "All points used in the validation"

Figure 3a: I think it is not fair comparison between SYS4 and SEAM when all start dates in SEAM are considered. Firstly, the sample sizes of both forecasts are not same, then more importantly the target dates of given lead time are not same.

R: Yes, we agree with that, and we have discussed that in the text and also shown more fair comparisons.

Page 7, Line 215-219: Not sure what the authors want to say "An explanation can …. driving forecast is used."

R: This was further explained: *"An explanation can be that the ensembles from the two meteorological forecasts are not matched member by member in terms of their relative deviation from the mean, for example matching members from each distribution according to their wetness"*

Figure 4 is hard to understand. "The dimension of the circles is proportional to the number of days while the color scale refers to progressive weeks". Number of days of what?

R: The colours and the circle size show the same thing, the colours are added to make the figure easier to understand. The figure text was changed to :

*"The number of weeks (days) before the CRPSS goes below zero using only the first forecast of the month for a) SEAM against CLIM; b) SYS4 against CLIM c) SEAM against SYS4; and d) difference between SEAM against CLIM and SYS4 against CLIM. The dimension of the circles is proportional to the number of days while the color scale refers to the number of weeks. The size and colour of the circles are therefore showing the same information and are both added for clarity."*

Page 8, Line 228: effect?

R: Changed to "effect"

Page 8, Line 236-240: Please rephrase this paragraph. What do authors mean by recent "development of the precipitation forecast?"

R: Precipitation forecast was changed to "precipitation model scheme"

Figure 5 caption: replace "functionality" with "function"

R: Changed

Figure 5: Legend for SYS4 is wrong (see previous comments)

R: The legend was corrected

Figure 7: In the previous figures, authors have shown 10 and 90 percentile, why they chose 25 and 75 percentiles. How are the percentiles computed for reliability diagrams?

R: The percentiles are computed as the probabilities of exceeding (subceeding) the 75 (25) percentiles. The figure will be changed to be more consistent with the rest of the paper.

Page 8, Line 252: "Both forecast systems are over-confident". How? For forecast probability less than 0.5, it is lower than the observed frequency, while for higher than 0.5, it is higher than the observed frequency.

R: For low probabilities, the forecast is not predicting the event often enough, and for the higher end of the probabilities the event is predicted too often, which means that the forecast is over-confident.

However, Figure 7 will be reworked to better reflect the reliabilities.